# YOLO-IR-Free: An Improved Algorithm for Real-Time Detection of Vehicles in Infrared Images

**DOI:** 10.3390/s23218723

**Published:** 2023-10-26

**Authors:** Zixuan Zhang, Jiong Huang, Gawen Hei, Wei Wang

**Affiliations:** 1College of Automation, Nanjing University of Information Science & Technology, Nanjing 210044, China; 2Business School, The Chinese University of Hong Kong, Hong Kong 999077, China; 3School of Physics, Mathematics and Computing, The University of Western Australia, Crawley, WA 6009, Australia; 4Jiangsu Collaborative Innovation Center of Atmospheric Environment and Equipment Technology (CICAEET), Nanjing University of Information Science & Technology, Nanjing 210044, China

**Keywords:** vehicle detection, YOLOv7, infrared, anchor-free

## Abstract

In the field of object detection algorithms, the task of infrared vehicle detection holds significant importance. By utilizing infrared sensors, this approach detects the thermal radiation emitted by vehicles, enabling robust vehicle detection even during nighttime or adverse weather conditions, thus enhancing traffic safety and the efficiency of intelligent driving systems. Current techniques for infrared vehicle detection encounter difficulties in handling low contrast, detecting small objects, and ensuring real-time performance. In the domain of lightweight object detection algorithms, certain existing methodologies face challenges in effectively balancing detection speed and accuracy for this specific task. In order to address this quandary, this paper presents an improved algorithm, called YOLO-IR-Free, an anchor-free approach based on improved attention mechanism YOLOv7 algorithm for real-time detection of infrared vehicles, to tackle these issues. We introduce a new attention mechanism and network module to effectively capture subtle textures and low-contrast features in infrared images. The use of an anchor-free detection head instead of an anchor-based detection head is employed to enhance detection speed. Experimental results demonstrate that YOLO-IR-Free outperforms other methods in terms of accuracy, recall rate, and average precision scores, while maintaining good real-time performance.

## 1. Introduction

The rapid development of computer vision and deep learning technologies [1,2,3] has brought revolutionary changes to the field of vehicle detection [4,5,6]. Deep learning, in particular, has made significant contributions to vehicle detection, especially in complex environments. Infrared images offer notable advantages in challenging conditions such as low light, haze, and nighttime, which can improve detection accuracy [7,8,9]. Applying deep learning to infrared image-based vehicle detection holds immense potential for intelligent transportation and autonomous driving.

However, current infrared vehicle detection technology faces several challenges within the domain of object detection algorithms, including issues related to low contrast, small object detection, and real-time performance. Some existing lightweight target detection algorithms struggle to effectively balance detection speed and accuracy in this particular task. In response to this challenge, researchers are actively exploring novel techniques and methodologies. One potential solution involves the integration of deep learning with conventional computer vision technologies to overcome the challenges associated with low contrast and small object detection. Furthermore, enhancing real-time performance is of paramount importance, particularly in the context of autonomous driving systems, as it necessitates rapid and precise detection of surrounding vehicles and obstacles. Infrared image-based vehicle detection holds significant value in various application scenarios, such as intelligent transportation and autonomous driving, making real-time performance and accuracy crucial requirements [10,11]. Current methods for infrared image-based vehicle detection demonstrate certain limitations, as feature-based approaches are susceptible to noise interference, and traditional machine learning methods struggle with handling complex environmental variations [11]. Although advanced generic object detection methods such as YOLOv7 [12] have achieved notable success in many domains, they have not been optimized specifically for the unique challenges and real-time requirements of infrared image-based vehicle detection. Therefore, it is necessary to explore new approaches to enhance the performance of infrared image-based vehicle detection. The motivation behind this research is to investigate new methods and optimization strategies to improve the performance of infrared image-based vehicle detection, fulfilling the requirements of real-time performance and accuracy.

The main contribution of this paper lies in the proposal of an improved algorithm, YOLO-IR-Free, for infrared vehicle detection. Specifically, the following points are highlighted:(1)We introduce YOLO-IR-Free, an anchor-free-based algorithm, aimed at fulfilling the real-time detection requirements in infrared images.(2)To address the challenge of capturing vehicle features in complex environments, we incorporate novel attention mechanisms and network modules to enhance the model’s capturing capabilities.(3)Through extensive experimental validation, our YOLO-IR-Free algorithm demonstrates outstanding performance in key performance metrics such as accuracy, recall, and F1 score. When compared to existing methods for infrared vehicle detection, our algorithm significantly improves detection performance.

These innovative contributions will propel the advancement of infrared vehicle detection technology and provide robust technical support for real-time infrared detection applications.

The structure and organization of this paper are as follows: In the Section 1, we elaborate on the research background, motivation, and key contributions. The Section 2 covers related work, providing a detailed review of the development in infrared image-based vehicle detection and existing methods, along with an analysis of their real-time performance and limitations. The Section 3 presents a detailed description of our proposed YOLO-IR-Free method, including the anchor-free detection head, the new attention mechanism, and network module. The Section 4 comprises experiments and evaluations, outlining the experimental setup, dataset, evaluation metrics, and experimental environment. We compare the performance of YOLO-IR-Free with other methods in terms of accuracy, recall rate, F1 score, and analyze the effectiveness of the proposed attention mechanism and network module. Finally, in the Section 5, we summarize the main research findings and innovations of this paper, indicate potential limitations of the proposed method, and suggest future research directions.

## 2. Related Work

This section provides a comprehensive review of the current research status and trends in the field of infrared image-based vehicle detection. It includes the application of deep learning methods, improvements in the YOLO series [12,13,14,15,16,17], anchor-free detection methods, and the application of attention mechanisms in object detection.

The development of infrared image-based vehicle detection technology has gone through stages from feature extraction to deep learning-based approaches [18,19]. Early methods primarily relied on handcrafted features such as edge detection and HOG features, followed by pattern recognition and machine learning techniques for classification. With the rise of deep learning, convolutional neural networks (CNNs) have gradually become the mainstream approach for infrared image-based vehicle detection [18,19]. Deep learning-based methods have achieved significant performance improvements. Typical deep learning methods include CNNs, R-CNN series [20], SSD [21], and YOLO series. These methods have addressed challenges such as low contrast and small object detection to varying degrees, but still exhibit certain limitations in terms of performance.

The YOLO series methods have gained significant attention in the field of object detection due to their efficiency in real-time applications and relatively high accuracy. In the task of infrared image-based vehicle detection, researchers have made targeted improvements to the YOLO series methods, optimizing them for infrared image characteristics and specific adjustments for vehicle detection tasks. However, existing improvement methods still have certain limitations in handling infrared vehicle detection in complex scenes.

Algorithms based on anchor-free detection heads abandon the concept of predefined anchor boxes and achieve object detection by directly predicting the key points or center points of the targets. Typical algorithms based on anchor-free detection heads include FOCS [22], CornerNet [23], CenterNet [24], and others. In FCOS, object detection is achieved by classifying and regressing each pixel. For each pixel, FCOS predicts whether it belongs to an object, the centerness of the object (indicating if the pixel is the center of the object), and the coordinates of the bounding box. Compared to methods based on anchor-based detection heads, FCOS does not require the generation of anchor boxes, thereby avoiding the process of anchor selection and adjustment, simplifying the object detection pipeline. CornerNet is a corner-based object detection method that aims to locate and classify objects by detecting their corners. It introduces a specialized network architecture that transforms the object detection problem into a key point detection problem. Specifically, CornerNet represents each object as two corners on the diagonal, namely the top-left corner and the bottom-right corner, and then utilizes a convolutional neural network (CNN) to detect these two corners. CenterNet is a center point-based object detection method that achieves object localization and classification by detecting the center points of objects. It proposes an end-to-end network architecture that can directly predict the center points, sizes, and categories of objects from the input image. Methods based on anchor-free detection heads demonstrate certain advantages in object detection tasks, such as simpler design and higher detection accuracy.

Attention mechanisms have achieved significant success in computer vision tasks. In the context of object detection, several methods have integrated attention mechanisms, such as Squeeze-and-Excitation Network (SENet) [25], Convolutional Block Attention Module (CBAM) [26], and others [27]. SENet introduces an operation called “Squeeze-and-Excitation” to dynamically adjust the weights of feature maps in each channel. It enables adaptive learning of the weights for each channel, allowing the network to focus more on important features and enhance model performance. The channel attention mechanism adjusts the weights of each channel by learning the correlation between channels, enabling the network to better utilize inter-channel information. The spatial attention mechanism, on the other hand, performs maximum and average pooling on feature maps along the spatial dimension and learns weights to fuse these two pooling results, capturing both global and local contextual information. By combining channel and spatial attention mechanisms, CBAM can finely adjust the weights of feature maps, enhancing the model’s perception of key features. These methods leverage attention mechanisms to focus on important regions and features in the image, thereby improving detection performance. The application of attention mechanisms in computer vision tasks has proven to be remarkably successful.

In 2022, Jiang et al. proposed a framework for object detection in UAV thermal infrared (TIR) images and videos using YOLO models [28]. The study utilizes convolutional neural network (CNN) architecture to extract features from TIR data captured by FLIR cameras. The results show high mean average precision (mAP) and fast detection speed with the YOLOv5-s model. It provides valuable insights into the qualitative and quantitative evaluation of object detection from TIR images and videos using deep learning models. In 2021, Du et al. presented a new approach to detect weak and occluded vehicles in complex infrared environments using an improved YOLOv4 model [29]. The proposed method involves secondary transfer learning from a visible dataset to an infrared dataset, and the addition of a hard negative example mining block to the YOLOv4 model. The results show that the improved model achieved a detection accuracy of 91.59%, indicating its potential for real-world applications in surveillance and security. In 2021, Liu presented a robust thermal infrared vehicle and pedestrian detection method for accurately detecting motion-blurred, tiny, and dense objects in complex scenes [30]. The method proposes an optimized feature selective anchor-free (FSAF) module with a weight parameter β, enhancing the detection performance of motion-blurred objects and improving detection precision for tiny and dense objects when combined with the YOLOv3 single-shot detector. Experimental results show that the proposed method outperforms other thermal infrared detection algorithms, achieving a mean average precision (mAP) of 72.2%. Existing methods for infrared image-based vehicle detection have made certain achievements in accuracy, but still have limitations. For example, some methods are limited in handling complex scenes, small objects, and low contrast, making them unable to meet the real-time detection requirements of edge computing platforms. On the other hand, while the YOLO series methods perform well in real-time applications, their accuracy and recall rates may be slightly inferior to other methods in certain cases. To address these issues, this paper proposes an improved method for infrared image-based vehicle detection to enhance detection performance.

## 3. Method

This section presents a detailed description of our proposed YOLO-IR-Free method, which includes improvements to the detection head through an anchor-free approach, a new attention mechanism, and network modules. Firstly, we introduce the basic structure and principles of the anchor-based detection head in the YOLO series algorithms and explain the motivation and methods to improve it into an anchor-free detection head. Next, we discuss the limitations of existing attention mechanisms and elaborate on the design and principles of the new attention mechanism. Finally, we analyze the deficiencies of the ResNet [31] residual module in infrared vehicle detection and propose the design ideas and implementation details of a new network module.

### 3.1. Anchor-Free Detection Head

Anchor-based detection heads rely on predefined anchor boxes of different sizes and aspect ratios to predict object locations and sizes within an image. These anchors serve as reference points for the model to learn how to adjust and refine object predictions based on the anchor boxes, making them suitable for handling a wide range of object sizes and shapes. In contrast, anchor-free detection heads do not rely on predefined anchors. Instead, they directly predict object bounding boxes by regressing the coordinates of object centers and their sizes, making them more flexible and efficient in object detection tasks. Anchor-free methods often excel in detecting objects with varying scales and orientations, simplifying the model architecture.

The YOLO series algorithms adopt an anchor-based detection head, which predicts the offsets and confidences for a predefined set of anchor points to achieve object detection. While this approach improves detection speed and accuracy to some extent, it still has limitations due to predefined anchor sizes and ratios, which may lead to poor detection performance for objects of different shapes and sizes.

To overcome the limitations of the anchor-based approach, we improve the original YOLOv7 detection head into an anchor-free method. Specifically, we abandon the concept of predefined anchor points and directly predict the center point and bounding box parameters of the targets for vehicle detection. This method handles objects of different shapes and sizes more flexibly, avoids the constraints of predefined anchor sizes and ratios, reduces computational and memory requirements, and improves detection speed. Compared to the anchor-based detection head, the design of the anchor-free detection head is simplified, as it does not require predefined anchor boxes, and it is more robust to variations in target scales and aspect ratios. The anchor-free method eliminates the need for generating a large number of candidate boxes at different scales and aspect ratios, reducing the box generation steps in the object detection process and improving detection speed. In contrast, the anchor-based method requires position offset and class prediction for each anchor box, resulting in higher computational complexity.

### 3.2. Introducing a New Attention Mechanism

Although existing attention mechanisms have achieved significant success in computer vision tasks, they still exhibit performance limitations in the domain of infrared image vehicle detection, particularly when handling complex scenes, small objects, and low contrast issues.

To enhance the performance of infrared image vehicle detection, we propose a new attention mechanism called the Thermal Spatial Excitation (TSE) module. This attention mechanism improves detection accuracy by focusing on important regions and features in the image. Specifically, we design a parallel attention mechanism with residual connections, which enhances the expressive power of features and allows the model to better preserve the information from the original features. Through residual connections, the model can better adapt to the attention-weighted features and avoid information loss. This improvement enhances the performance and feature representation capability of the model without significantly increasing computational complexity.

Figure 1 illustrates the proposed TSE module. The input features pass through two sets of SE channel attention mechanisms in parallel, along with a residual connection. The outputs of the two sets of SE channel attention mechanisms are concatenated with the residual output. Finally, a convolutional layer is applied to integrate the channel numbers, ensuring that the input and output shapes of the entire module are the same for easy integration. The SE channel attention mechanism uses global average pooling in the compression ($F_{sq}$) stage, which is applied to the input feature map, compressing it into a vector. This vector represents the global statistical information for each channel, which corresponds to the importance weight of each channel. In the excitation ($F_{ex}$) stage, the compressed vector is processed through a fully connected layer (typically a multi-layer perceptron). The purpose of this fully connected layer is to learn the weights for each channel, enhancing or suppressing the representational capability of different channels. Finally, in the scaling ($F_{scale}$) stage, the learned channel weights are multiplied with the original feature map, resulting in a weighted feature map. This weighted feature map highlights important channels and weakens unimportant ones, thereby improving feature discriminability.

Our TSE attention mechanism enhances feature representation and model expressiveness. By designing two sets of attention modules in parallel and concatenating their outputs, we introduce multiple attention mechanisms to capture different aspects of the input features. Each attention module can adaptively learn the relationship between channels and weight the feature maps, thereby enhancing the model’s focus on important information. The connected parallel attention modules further enhance feature representation, enabling the model to better capture the richness of input data. The concatenation of the outputs of the two attention modules with the residual connection implements a skip connection, which helps propagate information from lower to higher layers, avoiding information loss and degradation. This connection enables the model to better learn abstract and semantically meaningful features, enhancing its ability to capture details and global information.

### 3.3. Introducing New Network Modules

While the residual modules in ResNet have achieved significant success in many computer vision tasks, their performance is limited in infrared image vehicle detection. The main reason is that the ResNet residual modules have insufficient capturing capability for subtle textures and low-contrast features in infrared images, leading to suboptimal detection results.

To overcome the limitations of ResNet residual modules, we propose a new network module called Rep-LAN (Representation Learning with Attention Neurons). This module is designed to capture richer feature information and contextual information, enhancing the robustness of the network. Specifically, we design an efficient network module that controls the fusion of outputs from different gradient paths and combines the concept of RepConv (Representation Convolution) [32]. This allows the network to learn more diverse features and reduce the parameter count to some extent during inference. Figure 2 illustrates the proposed Rep-LAN network module, which consists of CBS (Convolutional Block [33] with Silu activation [34]) and RepConv. CBS is composed of a convolutional layer, batch normalization layer, and SiLU (Sigmoid Linear Unit) activation function. RepConv (shown in Figure 3) is a convolutional neural network structure designed for image recognition tasks. It enhances the feature representation capability of the network by introducing representation transformations and convolution operations within each convolutional layer. It can better capture local details and global contextual information, providing more powerful feature representation.

Our Rep-LAN combines the advantages of the ResNet residual structure, RepConv, and multi-gradient fusion. The skip connections allow information to directly propagate through the network, avoiding the problem of diminishing or amplifying gradients in deep layers. With the presence of skip connections, the network becomes easier to train and optimize, allowing for the construction of very deep networks that achieve better performance. Meanwhile, the RepConv structure enables the network to learn different feature representations within each convolutional layer, enhancing its ability to capture multi-scale and multi-level features. Through multi-gradient fusion, the network can effectively integrate gradient information from different levels, fully utilizing the multi-level feature representation capability. By applying these improvements to the YOLO-IR-Free method, we achieve high performance in infrared vehicle detection tasks.

### 3.4. YOLO-IR-Free

Before introducing our YOLO-IR-Free algorithm, it is imperative to acquaint oneself with several other lightweight object detection algorithms. YOLOv4-tiny [16], a streamlined variant of YOLOv4 [16], merits particular attention due to its parsimonious parameterization, merely encompassing 6 million parameters, equivalent to one-tenth of its precursor, which leads to a substantial acceleration in detection speed. Comprising a network architecture with a total depth of 38 layers, it incorporates three residual units, LeakyReLU activation functions, and a shift in target classification and regression to two distinct feature layers, synergistically leveraging a Feature Pyramid Network (FPN). YOLOv4-mobilenet [16,35], an offspring of YOLOv4 [16], diverges by adopting the lighter Mobilenet backbone. YOLOv5S [17], as developed by Glenn Jocher, represents another noteworthy lightweight object detection algorithm. In contrast to YOLOv4 [16], YOLOv5 [17] introduces an array of enhancements, including an enriched data augmentation regimen and Weighted NMS, among others. YOLOv7-Tiny [12], designed expressly for resource-constrained devices, stands as an efficient lightweight object detection algorithm. Building upon the foundation of YOLOv7, it achieves real-time operation on embedded and mobile platforms through a more compact network architecture and optimized training strategies, thereby reducing model parameters and computational requirements.

Figure 4 presents our proposed YOLO-IR-Free algorithm, which consists of the anchor-free head, TSE feature fusion mechanism module, and Rep-LAN network structure. The CBS module is the convolution layer, batch normalization layer, and SILU activation layer. The MP module is the maxpooling layer. The SILU activation function is utilized in the YOLO-IR-Free algorithm. Relative to several other lightweight object detection algorithms, our YOLO-IR-Free algorithm demonstrates a distinct advantage in the domain of infrared vehicle detection. Infrared vehicle images often present challenges such as occlusion and low contrast. Our TSE (Thermal Spatial Enhancement) attention mechanism enhances the focus on vehicle-related features, mitigating these challenges. Furthermore, the Rep-LAN (Receptive Field Pyramidal Long-range Attention Network) architecture amalgamates the computational efficiency of Rep-Conv (Receptive Field Pyramidal Convolution) with the benefits of short-path multi-block stacking, thereby augmenting the network’s feature extraction capabilities.

In terms of the loss function, we adopt the *CIoU* [36]. Compared to the traditional Intersection over Union (*IoU*) loss function, the *CIoU* loss function considers the differences in the position (Figure 5), scale, and shape of the predicted bounding boxes, thereby better optimizing the accuracy of the detection model. The formula for computing the *CIoU* loss function is defined as:(1)CIoU=IoU−ρ−δ−αv

Here, the center coordinates of the predicted bounding box and the ground truth bounding box are denoted as $\left(x,y\right)$ and $\left(\hat{x},\hat{y}\right)$, respectively, while the widths and heights are represented by w,h and $\hat{w},\hat{h}$. The conventional Intersection over Union (*IoU*, Figure 6) is defined as:(2)$$IoU\;=\;\frac{\mathrm{Intersection}}{\mathrm{Union}}\;=\;\frac{A\;\cap \;\hat{A}}{A\;\cup \hat{A}}$$
where *A* represents the area of the box, A=wh, and Ā represents the area of the predicted box, *Ā* = $\hat{A}=\hat{w\hat{h}}$. In order to enhance performance, the *CIoU* loss function further introduces the size difference δ and the relative area difference ρ:(3)$$\delta =\frac{\hat{d}-d}{\hat{d}}$$
(4)$$\rho =\frac{\hat{A}-A}{\hat{A}}$$

Here, d and $\hat{d}$ represent the diagonal lengths of the two boxes, and the offset of the center points between the predicted box and the ground truth box is denoted as $p=\left(\hat{x}-x,\hat{y}-y\right)$. To penalize the differences in aspect ratio and balance the importance of aspect ratio and position, the *CIoU* loss function introduces adjustment coefficients v and α:(5)$$v=\left(4/\pi^{2}\right)\times {\left(\arctan\left(w/h\right)-\arctan\left(\hat{w}/\hat{h}\right)\right)}^{2}\alpha$$
(6)$$\alpha =\frac{v}{\left(1-IoU\right)+v}$$

By utilizing the *CIoU* loss function, it becomes possible to more accurately measure the differences between the predicted bounding boxes and the ground truth boxes, and to model the differences in position, scale, and shape more precisely, thereby enhancing the performance of the object detection model.

## 4. Experimental Results and Discussion

In this section, we assess the performance of the YOLO-IR-Free method. Firstly, we introduce the experimental dataset, evaluation metrics, and experimental environment. Then, we conduct a comparative analysis of YOLO-IR-Free with other methods in terms of real-time capability and performance. Finally, we validate the effectiveness of the proposed attention mechanism and network module in enhancing the model’s performance.

### 4.1. Experimental Setup

To evaluate the performance of YOLO-IR-Free, we selected an infrared vehicle detection dataset that encompasses various scenarios and environmental conditions. Figure 7 depicts a subset of the utilized infrared vehicle image dataset in this study, which comprises infrared images captured from multiple perspectives by an unmanned aerial vehicle equipped with an infrared camera. The dataset encompasses instances of occlusion, including obstructions such as trees along the roadside and various buildings. The vehicular instances within the dataset encompass elongated cargo vehicles as well as compact sedans, exhibiting substantial variation in dimensions and aspect ratios. Notably, certain images within the dataset exhibit varying degrees of blurriness. The presence of these challenging attributes collectively imposes a rigorous evaluation of the detection algorithm’s performance. The dataset consists of a total of 13,780 infrared images, with 11,500 images used for training and 2280 images for testing. The input size of the images is set to 320 × 320.

We adopt precision, recall, F1 score, and mean average precision (mAP) as evaluation metrics to quantify the performance of different methods. The experiments are conducted on a computer equipped with an NVIDIA RTX 3090 graphics card (Nvidia, Santa Clara, CA, USA), Intel Core i5-13600KF processor (Intel, Santa Clara, CA, USA), and 32 GB of memory. All methods utilize the same hardware and software environment.

### 4.2. Comparison of YOLO-IR-Free with Other Methods

To demonstrate the performance advantages of YOLO-IR-Free, we compare it with the original YOLOv7 and other improved algorithms. Table 1 presents the performance comparison between our YOLO-IR-Free algorithm and Faster-RCNN [37], YOLOv4-mobilenet, YOLOv4-tiny, YOLOv5s, and YOLOv7-tiny. In terms of mean average precision (mAP), our YOLO-IR-Free achieves the highest value of 0.95 among all algorithms. The second and third-ranked algorithms are YOLOv5s with a precision of 0.925 and YOLOv7-tiny with a precision of 0.923. For recall, our YOLO-IR-Free achieves the highest value of 0.902, which is also the best among all compared algorithms. In terms of F1 score, YOLO-IR-Free achieves 0.92, while YOLOv7-tiny and YOLOv5s score 0.88 and 0.89, respectively. Furthermore, YOLO-IR-Free exhibits outstanding real-time capability, reaching a frame rate of 192 FPS, far surpassing Faster-RCNN, YOLOv4-mobilenet, and YOLOv5s. The YOLOv7-tiny algorithm achieves a higher frames per second (FPS) compared to YOLOv8s [38], yet exhibits lower values across key performance metrics including precision, recall, F1, and mean average precision (mAP) in comparison to YOLOv8s [38]. In contrast, our YOLO-IR-Free algorithm demonstrates notable improvements over YOLOv7-tiny, boasting a slight advantage over YOLOv8s [38] in terms of precision, recall, and mAP. Moreover, it holds a significant edge in terms of FPS. Despite YOLOv4-tiny having the highest FPS, its detection accuracy is significantly lower.

Figure 8 illustrates the F1 score curve at various confidence threshold levels. Our YOLO-IR-Free algorithm consistently outperforms other algorithms, maintaining the highest F1 score from the beginning to a confidence threshold of 0.75. In contrast, the Faster-RCNN algorithm achieves the lowest F1 score. These results demonstrate the clear advantages of YOLO-IR-Free in terms of both performance and real-time capability.

We present a comprehensive data analysis based on the precision and recall performance metrics of state-of-the-art object detection algorithms, including YOLOv7-Tiny, YOLOv5s, YOLOv4-Tiny, YOLOv4-MobileNet, and our proposed YOLO-IR-Free algorithm. The analysis is conducted using precision-recall curves plotted against varying confidence threshold values. The primary focus is on evaluating the effectiveness of the proposed YOLO-IR-Free algorithm, which incorporates a novel TSE (Thermal Spatial Excitation) attention mechanism to enhance feature representation and object localization. The curves in Figure 8 and Figure 9 vividly illustrate the algorithmic comparisons across different confidence threshold ranges.

In Figure 9, the precision curves demonstrate the performance of the aforementioned algorithms across different confidence threshold levels (0 to 0.48). Notably, our YOLO-IR-Free algorithm consistently outperforms the other algorithms in terms of precision within this range. This is attributed to the integration of the TSE attention mechanism, which significantly enhances the model’s ability to capture intricate vehicle details. As a result, the precision achieved by YOLO-IR-Free is on par with YOLOv7-Tiny and YOLOv5s, showcasing its efficacy in vehicle detection tasks. Comparatively, Faster-RCNN algorithm exhibits the lowest precision scores, possibly due to its multi-scale feature fusion limitations. YOLOv4-Tiny initially closely follows YOLOv5s but subsequently falls behind YOLO-IR-Free, YOLOv5s, and YOLOv7-Tiny. Similarly, YOLOv4-MobileNet performs well in the early confidence threshold range but trails behind YOLO-IR-Free, YOLOv7-Tiny, YOLOv5s, and YOLOv4-Tiny.

Moving to Figure 10, the recall curves highlight the recall performance of the algorithms across varying confidence threshold values (0 to 0.75). Notably, the YOLO-IR-Free algorithm consistently achieves higher recall scores compared to the other algorithms within this range. This can be attributed to the adoption of an anchor-free detection head in our algorithm, facilitating the detection of objects with diverse size ratios. The curves for YOLOv5s and YOLOv7-Tiny maintain a consistent trajectory, reflecting their stable performance throughout. YOLOv4-Tiny, YOLOv4-MobileNet, and Faster-RCNN algorithms exhibit relatively smoother curves, but they initially lag behind the other algorithms in recall. The occasional higher recall curve of Faster-RCNN in comparison to YOLOv4-MobileNet indicates the advantageous impact of two-stage algorithms on recall rates. The precision and recall analysis presented in Figure 8 and Figure 9 showcases the superiority of the proposed YOLO-IR-Free algorithm in both precision and recall metrics. The integration of the TSE attention mechanism and the anchor-free detection head contribute significantly to its enhanced performance. YOLOv7-Tiny and YOLOv5s exhibit competitive precision and recall results, while YOLOv4-Tiny, YOLOv4-MobileNet, and Faster-RCNN demonstrate distinct strengths and weaknesses in precision and recall across various confidence threshold ranges. This analysis provides valuable insights into the comparative performance of these algorithms in vehicle detection tasks and serves as a foundation for further algorithmic enhancements in the field of object detection.

Figure 11 compares the performance of our algorithm with Faster-RCNN, YOLOv4-mobilenet, YOLOv4-tiny, YOLOv5s, and YOLOv7-tiny in terms of actual inference results. The yellow circles represent missed detections, while the blue circles indicate poorly localized bounding boxes. From the figure, we observe that Faster-RCNN, YOLOv4-mobilenet, and YOLOv4-tiny algorithms exhibit varying degrees of missed detections and low-quality bounding boxes. YOLOv5s algorithm has two missed detections. In the bottom right corner of the figure, a car is only partially visible, making it a challenging detection scenario. Both YOLOv7-tiny and our algorithm missed detecting this car. However, our algorithm demonstrates superior detection performance for all vehicles, outperforming the other algorithms significantly.

To demonstrate the performance advantages of YOLO-IR-Free, we compare it with the original YOLOv7 and other improved algorithms. Table 1 presents the performance comparison between our YOLO-IR-Free algorithm and Faster-RCNN.

### 4.3. Ablation Experiment

In order to validate the impact of the anchor-free detection head on network performance, we initially replaced the detection head of YOLO-IR-Free with a conventional anchor-based detection head for experimentation. Table 2 presents the performance comparison upon removal of the anchor-free detection head. It is observed that precision, recall, and mAP experience slight decreases, whereas FPS exhibits a significant decline. The experimental results underscore the enhancements brought about by the anchor-free detection head across various performance metrics, notably a substantial improvement in inference speed.

To validate the performance improvement brought about by the proposed attention mechanism and network module, we systematically remove the TSE module and Rep-LAN network structure from YOLO-IR-Free to evaluate the model’s performance.

Initially, we conduct experiments by removing the TSE attention module from the YOLO-IR-Free network. As is clear from Table 3, the results show a 1.3% decrease in mAP compared to the YOLO-IR-Free algorithm. Additionally, there is a certain degree of decline in F1 score, recall, and precision.

Furthermore, we proceed to remove the Rep-LAN network structure, resulting in an additional decrease of 3.1% in mAP. Similarly, there is a decrease in F1 score, recall, and precision. These findings confirm the significant role played by the proposed attention mechanism and network module in enhancing the model’s performance.

Overall, the results demonstrate the effectiveness of the attention mechanism and network module in improving the performance of the YOLO-IR-Free method.

### 4.4. Comparison with Other Infrared Vehicle Detection Algorithms

In this section, we provide a comprehensive performance comparison between our proposed YOLO-IR-Free algorithm and three other notable infrared vehicle detection algorithms. The first algorithm, developed by Liu et al. and published in the Sensors journal in 2021, is an improvement upon YOLOv3 tailored for infrared vehicle detection [30]. The second algorithm, presented by Du et al. in the IEEE ACCESS journal in 2021, builds upon YOLOv4 with enhancements for infrared vehicle detection [29]. The third algorithm, introduced by Jiang et al. in the International Journal of Applied Earth Observations and Geoinformation in 2022 (with an impressive Impact Factor of 7.5), adapts the YOLOv5s architecture for infrared vehicle detection [28].

Our experimental findings reveal that our YOLO-IR-Free algorithm consistently outperforms the aforementioned algorithms across several crucial performance metrics. While Du’s algorithm exhibits a marginal advantage in precision, our algorithm excels significantly in recall, F1 score, and mAP. Table 4 provides a detailed performance comparison against these state-of-the-art infrared vehicle detection algorithms.

In summary, our YOLO-IR-Free algorithm emerges as the leading choice for infrared vehicle detection, consistently surpassing other state-of-the-art algorithms in terms of mAP, and demonstrating substantial advantages in recall and F1 score while maintaining competitive precision. This underscores the efficacy and potential of our proposed approach in advancing the field of infrared object detection.

## 5. Conclusions

In conclusion, this paper has presented YOLO-IR-Free, an advanced algorithm tailored for the real-time detection of vehicles in infrared imagery. The motivation behind this work stemmed from the critical need to enhance traffic safety and the efficiency of intelligent driving systems by addressing the challenges associated with infrared vehicle detection. These challenges include low contrast, the detection of small objects, and the imperative requirement for real-time performance.

To overcome these challenges, we introduced a novel attention mechanism and network module into the YOLOv7 framework, effectively empowering our algorithm to capture subtle textures and low-contrast features inherent in infrared images. Notably, our decision to employ an anchor-free detection head, as opposed to the traditional anchor-based approach, significantly improved detection speed without compromising accuracy. Through extensive experiments, YOLO-IR-Free demonstrated superior performance metrics, including accuracy, recall rate, and F1 score, all while maintaining impeccable real-time capabilities.

Looking ahead, the future holds exciting prospects for YOLO-IR-Free. We envision further optimizations to enhance its overall performance and real-time capabilities. Additionally, we anticipate extending the application of our algorithm to tackle a broader spectrum of infrared object detection tasks. As intelligent driving systems continue to evolve, YOLO-IR-Free stands as a promising solution for safer and more efficient roadways, underpinning the potential for groundbreaking advancements in the field of infrared object detection.

## Figures and Tables

**Figure 1 sensors-23-08723-f001:**
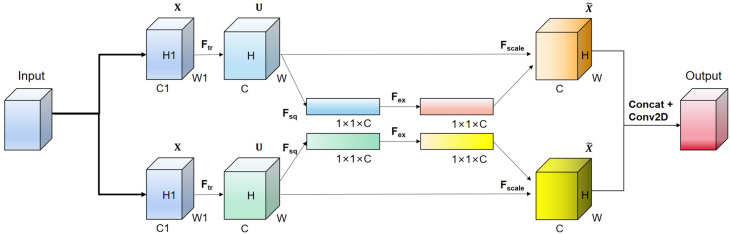
Our TSE attention mechanism.

**Figure 2 sensors-23-08723-f002:**
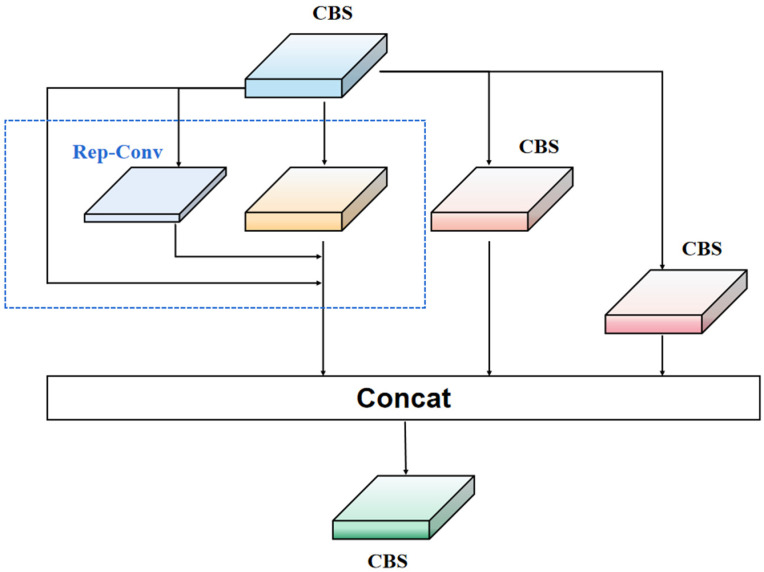
Rep-LAN module structure diagram.

**Figure 3 sensors-23-08723-f003:**
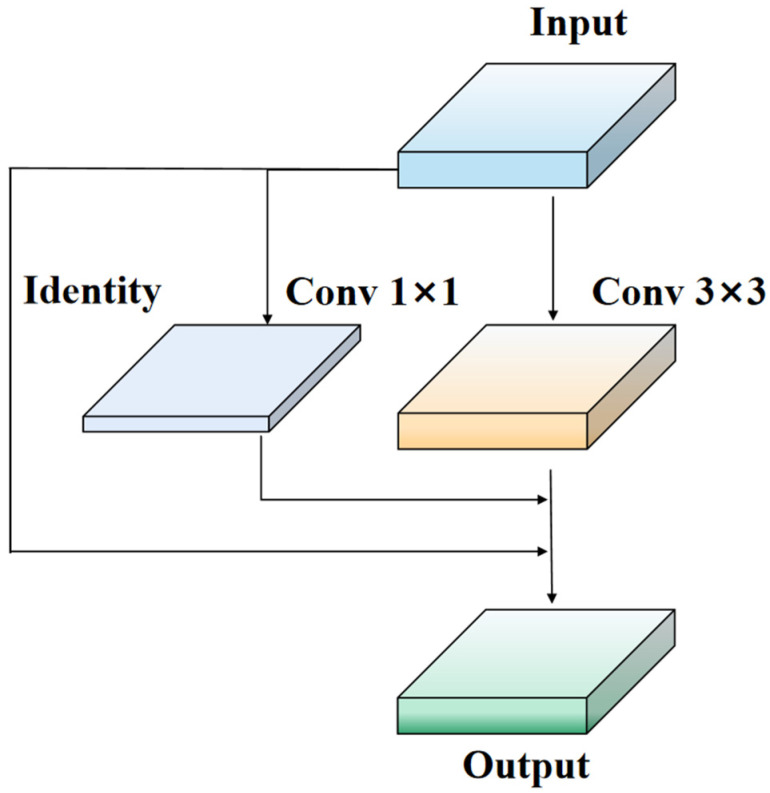
RepConv VGG module structure diagram.

**Figure 4 sensors-23-08723-f004:**
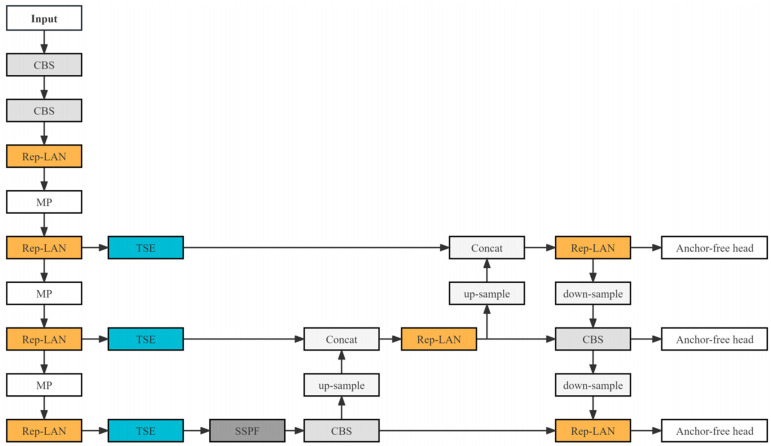
The network architecture diagram of YOLO-IR-Free.

**Figure 5 sensors-23-08723-f005:**
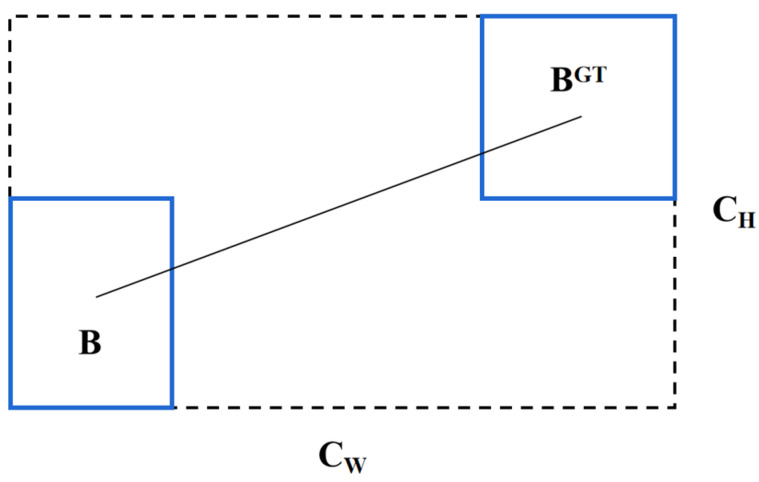
Calculate the distance from the GT box and the predicted box.

**Figure 6 sensors-23-08723-f006:**
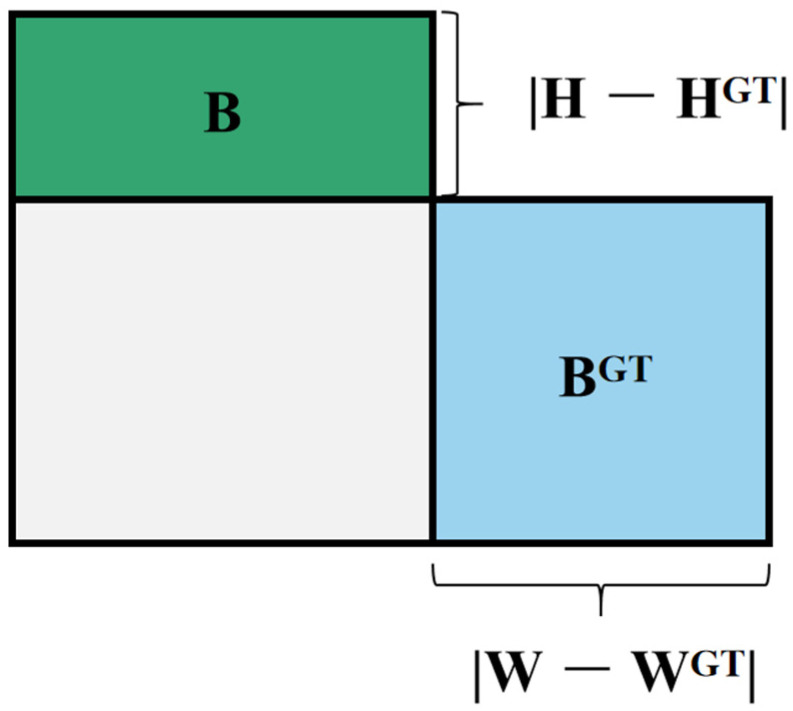
GT box and prediction box to calculate IoU.

**Figure 7 sensors-23-08723-f007:**
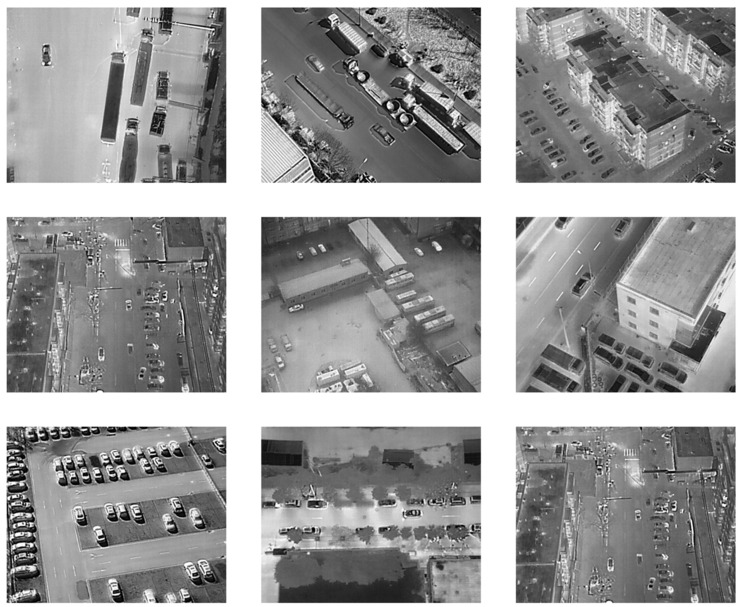
Part of the dataset used in this study.

**Figure 8 sensors-23-08723-f008:**
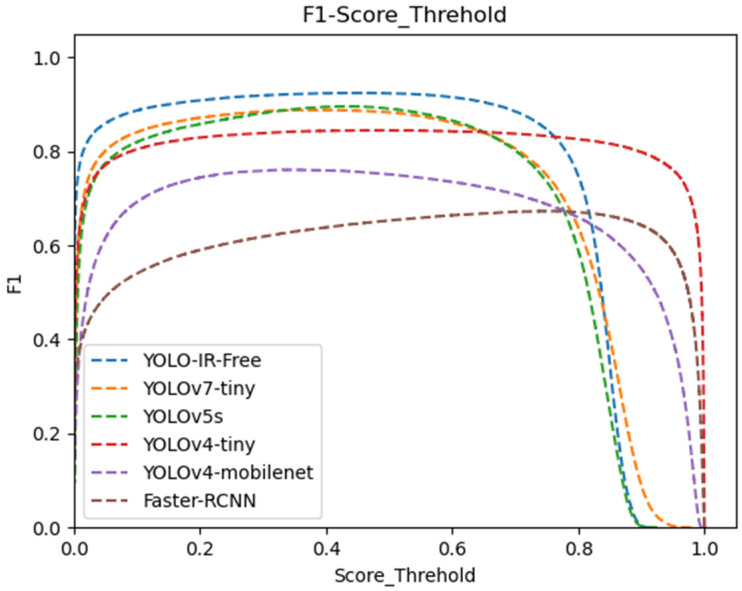
Confidence threshold vs. F1 score curves for YOLOv7-Tiny, YOLOv5s, YOLOv4-Tiny, YOLOv4-mobileNet, and our algorithm. The higher the curve, the better.

**Figure 9 sensors-23-08723-f009:**
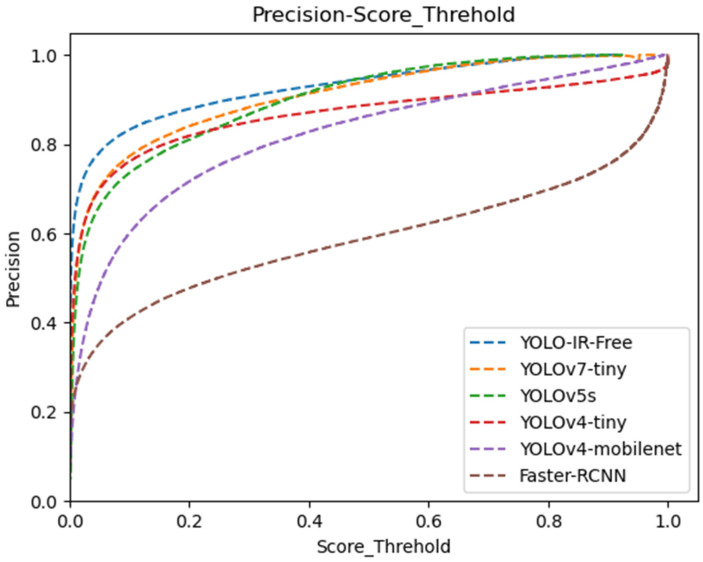
Confidence threshold vs. precision curves for YOLOv7-Tiny, YOLOv5s, YOLOv4-Tiny, YOLOv4-mobileNet, and our algorithm. The higher the curve, the better.

**Figure 10 sensors-23-08723-f010:**
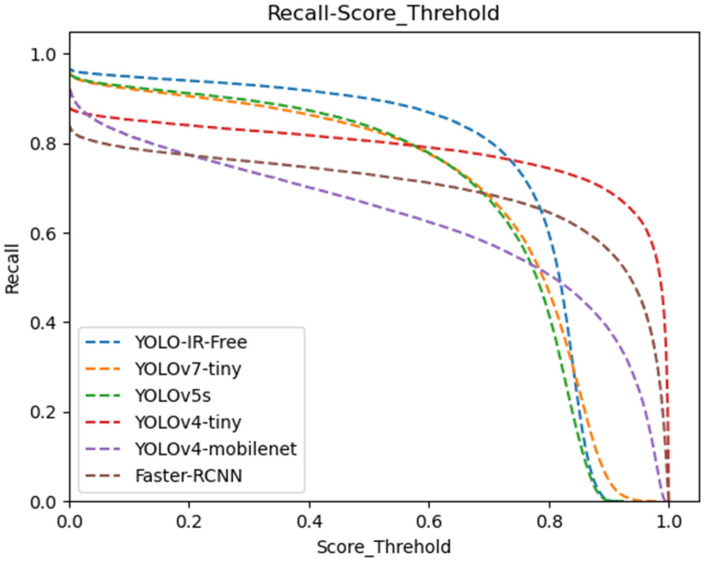
Confidence threshold vs. recall curves for YOLOv7-Tiny, YOLOv5s, YOLOv4-Tiny, YOLOv4-mobileNet, and our algorithm. The higher the curve, the better.

**Figure 11 sensors-23-08723-f011:**
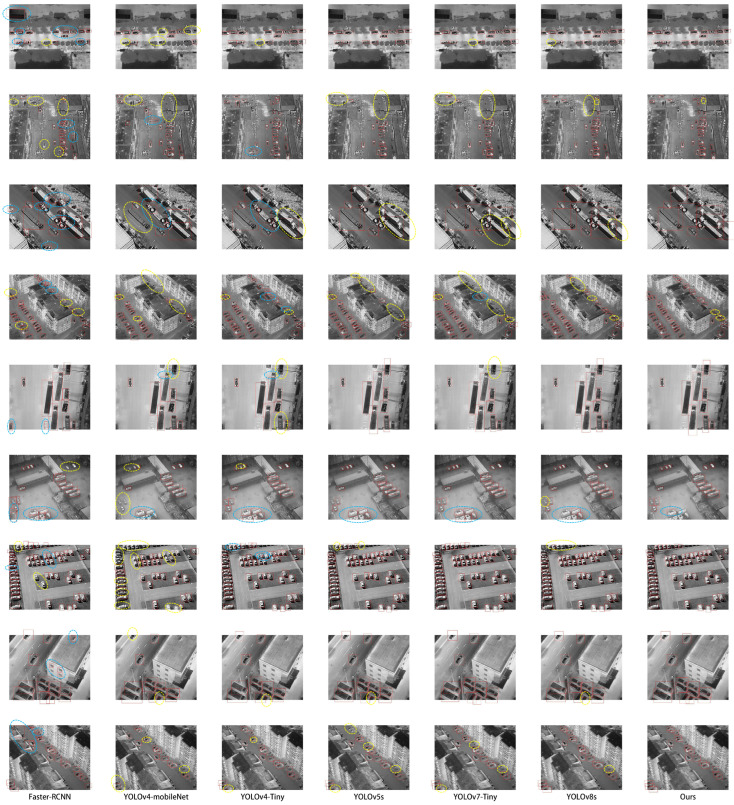
Comparison of detection results between Faster-RCNN, YOLOv4-mobileNet, YOLOv4-Tiny, YOLOv5s, YOLOv7-Tiny, and our algorithm. The red box indicates that the detector believes it is the correct target. The yellow dotted circle indicates that the detector missed detection, and the blue dotted circle indicates that the quality of the detection frame is too low.

**Table 1 sensors-23-08723-t001:** Performance evaluation comparison of different object detection algorithms.

Method	Precision (0.5)	Recall (0.5)	F1 (0.5)	mAP	FPS	ms
Faster-RCNN	0.59	0.73	0.65	0.654	25	40
YOLOv4- mobilenet	0.864	0.664	0.75	0.796	145	6.89
YOLOv4-tiny	0.888	0.805	0.84	0.832	243	4.11
YOLOv5s	0.951	0.837	0.89	0.925	131	7.63
YOLOv7-tiny	0.941	0.831	0.88	0.923	172	5.81
YOLOv8s	0.943	0.9	0.92	0.946	162	6.21
YOLO-IR-Free	0.945	0.902	0.92	0.950	192	5.2

**Table 2 sensors-23-08723-t002:** Performance comparison with anchor-free detection head removed.

Method	Precision (0.5)	Recall (0.5)	F1 (0.5)	mAP	FPS
Without Anchor-Free	0.943	0.9	0.92	0.946	167
With Anchor-Free	0.945	0.902	0.92	0.950	192

**Table 3 sensors-23-08723-t003:** Performance evaluation comparison of YOLO-IR-Free and YOLO-IR-Free without TSE or Rep-LAN module.

Method	Precision (0.5)	Recall (0.5)	F1 (0.5)	mAP
YOLO-IR-Free	0.945	0.902	0.92	0.950
Without TSE	0.931	0.889	0.91	0.937
Without Rep-LAN	0.905	0.875	0.89	0.906

**Table 4 sensors-23-08723-t004:** Performance comparison with state-of-the-art infrared vehicle detection algorithms.

Method	Precision (0.5)	Recall (0.5)	F1 (0.5)	mAP
Liu’s [30]	-	-	-	0.722
Du’s [29]	0.962	0.81	0.879	0.915
Jiang’s [28]	0.729	0.766	-	0.918
Ours	0.945	0.902	0.92	0.950

## Data Availability

Not applicable.

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
