# Peer review of "YOLO-IR-Free: An Improved Algorithm for Real-Time Detection of Vehicles in Infrared Images"

_sensors, 2023, doi:10.3390/s23218723_

Round 1

Reviewer 1 Report

The paper under review addresses the significant challenge of real-time detection of vehicles in infrared images through its proposed algorithm, YOLO-IR-Free. However, it is essential to note that the claim of addressing limitations in existing methods for detecting low-contrast small objects requires further substantiation.

  1. While the paper introduces YOLO-IR-Free as an Anchor-Free approach based on the YOLOv7 algorithm, evaluating the novelty of this contribution within the existing landscape is crucial. Notably, there have already been well-established anchor-free variants of YOLO such as YOLOX, PP-YOLO, and YOLOv8. The paper's contribution seems more aligned with a contextual implementation of existing algorithms rather than a novel algorithm in its own right.

  2. The paper's presentation of results raises concerns regarding clarity and conclusiveness. The ambiguity of the results, coupled with the vague depiction in Figure 11 to substantiate the superiority of the proposed algorithm, calls for further elaboration and detailed performance evaluation. The use of simple scenarios for implementation limits the extrapolation of the algorithm's efficacy to more complex real-world scenarios.

  3. The paper needs to provide a more comprehensive comparison with state-of-the-art platforms in the field of real-time vehicle detection in infrared images. The availability of superior existing platforms that perform similar tasks questions the practical significance of the proposed YOLO-IR-Free algorithm.

  4. The assertion of achieving real-time performance remains ambiguous and needs to be substantiated with concrete evidence. The paper should provide a detailed analysis of its real-time capabilities and compare them with existing benchmarks.

  5. In evaluating the paper's contribution to the existing literature, it is evident that the methodology and application do not present any significant breakthroughs. The paper lacks originality both in terms of its algorithmic methodology and its application context.

Given the aforementioned observations, it is my recommendation that the paper be rejected for publication in the "Sensors" journal. The paper's contributions do not align with the level of novelty and rigor expected for publication in a reputable journal.

Author Response

Dear Editor/Reviewer,

Thank you for reviewing our previous submission and providing valuable feedback and suggestions. We are hereby submitting the revised manuscript, along with a detailed response addressing each of the review comments and outlining the modifications we have made.

Firstly, we sincerely appreciate your recognition of our research topic and objectives, as well as your encouragement and support for our work. In the revised manuscript, we have carefully considered each of the comments you raised and made the necessary revisions and improvements accordingly. Please refer to the "Responses" section for the specific details of the modifications, and we have highlighted the revised portions in the revised manuscript.

Lastly, we would like to express our gratitude once again for your attention and review of our research. We deeply appreciate the valuable insights you have provided, as they have played a crucial role in enhancing the quality and academic rigor of our study. We hope that you will be satisfied with our revised manuscript and consider it for publication in your esteemed journal.

Best regards,

Zixuan Zhang

[email protected]

Response

While the paper introduces YOLO-IR-Free as an Anchor-Free approach based on the YOLOv7 algorithm, evaluating the novelty of this contribution within the existing landscape is crucial. Notably, there have already been well-established anchor-free variants of YOLO such as YOLOX, PP-YOLO, and YOLOv8. The paper's contribution seems more aligned with a contextual implementation of existing algorithms rather than a novel algorithm in its own right.

Response: Thank you for your suggestion. During our research, YOLOv8 had not been released. In the revised manuscript, we have added comparative experiments with the latest YOLOv8 algorithm.

“The YOLOv7-tiny algorithm achieves a higher Frames Per Second (FPS) compared to YOLOv8s, yet exhibits lower values across key performance metrics including Precision, Recall, F1, and mean Average Precision (mAP) in comparison to YOLOv8s. In contrast, our YOLO-IR-Free algorithm demonstrates notable improvements over YOLOv7-tiny, boasting a slight advantage over YOLOv8s in terms of Precision, Recall, and mAP. Moreover, it holds a significant edge in terms of FPS.”

The paper's presentation of results raises concerns regarding clarity and conclusiveness. The ambiguity of the results, coupled with the vague depiction in Figure 11 to substantiate the superiority of the proposed algorithm, calls for further elaboration and detailed performance evaluation. The use of simple scenarios for implementation limits the extrapolation of the algorithm's efficacy to more complex real-world scenarios.

Response: Thank you for your comments. Figure 11 clearly demonstrates the differences in performance between our algorithm and several other algorithms. We extensively analyzed and compared the performance of our algorithm in comparative experiments and ablation experiments. Moreover, the revised manuscript includes comparative experiments with YOLOv8s and three other recent algorithms for infrared vehicle detection tasks. Regarding your mention of more complex scenarios, they are not within the scope of our research as we specifically focus on the task of infrared vehicle detection.

The paper needs to provide a more comprehensive comparison with state-of-the-art platforms in the field of real-time vehicle detection in infrared images. The availability of superior existing platforms that perform similar tasks questions the practical significance of the proposed YOLO-IR-Free algorithm.

Response: Thank you for the suggestion. We have added comparisons with recent state-of-the-art algorithms in the field of infrared vehicle detection in the experimental section. Additionally, we have included a comparison with the latest YOLOv8s algorithm.

“The YOLOv7-tiny algorithm achieves a higher Frames Per Second (FPS) compared to YOLOv8s, yet exhibits lower values across key performance metrics including Precision, Recall, F1, and mean Average Precision (mAP) in comparison to YOLOv8s. In contrast, our YOLO-IR-Free algorithm demonstrates notable improvements over YOLOv7-tiny, boasting a slight advantage over YOLOv8s in terms of Precision, Recall, and mAP. Moreover, it holds a significant edge in terms of FPS.”

“4.4 Comparison with Other Infrared Vehicle Detection Algorithms

In this section, we provide a comprehensive performance comparison between our proposed YOLO-IR-Free algorithm and three other notable infrared vehicle detection algorithms. The first algorithm, developed by Liu et al. and published in the Sensors journal in 2021, is an improvement upon YOLOv3 tailored for infrared vehicle detection. The second algorithm, presented by Du et al. in the IEEE ACCESS journal in 2021, builds upon YOLOv4 with enhancements for infrared vehicle detection. The third algorithm, introduced by Jiang et al. in the International Journal of Applied Earth Observations and Geoinformation in 2022 (with an impressive Impact Factor of 7.5), adapts the YOLOv5s architecture for infrared vehicle detection.

Our experimental findings reveal that our YOLO-IR-Free algorithm consistently outperforms the aforementioned algorithms across several crucial performance metrics. While Du's algorithm exhibits a marginal advantage in Precision, our algorithm excels significantly in Recall, F1 score, and mAP. Table 4 provides a detailed performance comparison against these state-of-the-art infrared vehicle detection algorithms.

In summary, our YOLO-IR-Free algorithm emerges as the leading choice for infrared vehicle detection, consistently surpassing other state-of-the-art algorithms in terms of mAP, and demonstrating substantial advantages in Recall, F1 score, while maintaining competitive Precision. This underscores the efficacy and potential of our proposed approach in advancing the field of infrared object detection.”

The assertion of achieving real-time performance remains ambiguous and needs to be substantiated with concrete evidence. The paper should provide a detailed analysis of its real-time capabilities and compare them with existing benchmarks.

Response: Thank you for your comments. We have compared the algorithms' Frames Per Second (FPS) in Table 1. FPS is a commonly used metric in the field of object detection to assess real-time performance.

In evaluating the paper's contribution to the existing literature, it is evident that the methodology and application do not present any significant breakthroughs. The paper lacks originality both in terms of its algorithmic methodology and its application context.

Response: Thank you for your comments. We have proposed new attention mechanisms and network modules and further improved the Anchor-Free detection head based on YOLOv7. The final performance of our approach surpasses that of other algorithms.

Reviewer 2 Report

Based on the anchor-free detection method, this paper proposed an
enhanced algorithm, YOLO-IR-FREE, for real-time vehicle detection in
infrared images. It improves the attention mechanism and network module,
and the experimental results proved its performance.

1. Section 3.1, an appropriate description of the anchor-less detection
head should be added.

2. Some descriptions in Figure 1 are inconsistent with the main text:
"F_ez" is in the text, while "F_ex" is in Figure 1.

3. What does "MP" in Figure 4 mean?

4. Has it been compared with the most advanced (SOTA) methods?

5. The image is not clear when enlarged. 

Author Response

Dear Editor/Reviewer,

Thank you for reviewing our previous submission and providing valuable feedback and suggestions. We are hereby submitting the revised manuscript, along with a detailed response addressing each of the review comments and outlining the modifications we have made.

Firstly, we sincerely appreciate your recognition of our research topic and objectives, as well as your encouragement and support for our work. In the revised manuscript, we have carefully considered each of the comments you raised and made the necessary revisions and improvements accordingly. Please refer to the "Responses" section for the specific details of the modifications, and we have highlighted the revised portions in the revised manuscript.

Lastly, we would like to express our gratitude once again for your attention and review of our research. We deeply appreciate the valuable insights you have provided, as they have played a crucial role in enhancing the quality and academic rigor of our study. We hope that you will be satisfied with our revised manuscript and consider it for publication in your esteemed journal.

Best regards,

Zixuan Zhang

[email protected]

  1. Section 3.1, an appropriate description of the anchor-less detection head should be added.

Response: Thank you for your comments. I'm sorry, but we couldn't find information about the "anchor-less" detection head algorithm. We introduced anchor-free and anchor-based methods in Section 3.1.

“Anchor-based detection heads rely on predefined anchor boxes of different sizes and aspect ratios to predict object locations and sizes within an image. These anchors serve as reference points for the model to learn how to adjust and refine object predictions based on the anchor boxes, making them suitable for handling a wide range of object sizes and shapes.In contrast, anchor-free detection heads do not rely on predefined anchors. Instead, they directly predict object bounding boxes by regressing the coordinates of object centers and their sizes, making them more flexible and efficient in object detection tasks. Anchor-free methods often excel in detecting objects with varying scales and orientations, simplifying the model architecture.”

  1. Some descriptions in Figure 1 are inconsistent with the main text: "F_ez" is in the text, while "F_ex" is in Figure 1.

Response: Thank you for your careful review. I apologize for the error in our writing. The correct text should be "F_ex." We have made the correction.

  1. What does "MP" in Figure 4 mean?

Response: Thank you for your careful review. We have included an explanation of 'MP' (Maxpooling layer) in our revised manuscript. "The CBS module is the Convolution layer, Batch Normalization layer and SILU activation layer. The MP module is the Maxpooling layer. "

  1. Has it been compared with the most advanced (SOTA) methods?

Response: Thank you for the suggestion. We have added comparative experiments with YOLOv8s and, in Section 4.4, included a comparison with the state-of-the-art (SOTA) algorithms for infrared vehicle detection.

“The YOLOv7-tiny algorithm achieves a higher Frames Per Second (FPS) compared to YOLOv8s, yet exhibits lower values across key performance metrics including Precision, Recall, F1, and mean Average Precision (mAP) in comparison to YOLOv8s. In contrast, our YOLO-IR-Free algorithm demonstrates notable improvements over YOLOv7-tiny, boasting a slight advantage over YOLOv8s in terms of Precision, Recall, and mAP. Moreover, it holds a significant edge in terms of FPS.”

"4.4 Comparison with Other Infrared Vehicle Detection Algorithms

In this section, we provide a comprehensive performance comparison between our proposed YOLO-IR-Free algorithm and three other notable infrared vehicle detection algorithms. The first algorithm, developed by Liu et al. and published in the Sensors journal in 2021, is an improvement upon YOLOv3 tailored for infrared vehicle detection. The second algorithm, presented by Du et al. in the IEEE ACCESS journal in 2021, builds upon YOLOv4 with enhancements for infrared vehicle detection. The third algorithm, introduced by Jiang et al. in the International Journal of Applied Earth Observations and Geoinformation in 2022 (with an impressive Impact Factor of 7.5), adapts the YOLOv5s architecture for infrared vehicle detection.

Our experimental findings reveal that our YOLO-IR-Free algorithm consistently outperforms the aforementioned algorithms across several crucial performance metrics. While Du's algorithm exhibits a marginal advantage in Precision, our algorithm excels significantly in Recall, F1 score, and mAP. Table 4 provides a detailed performance comparison against these state-of-the-art infrared vehicle detection algorithms.

In summary, our YOLO-IR-Free algorithm emerges as the leading choice for infrared vehicle detection, consistently surpassing other state-of-the-art algorithms in terms of mAP, and demonstrating substantial advantages in Recall, F1 score, while maintaining competitive Precision. This underscores the efficacy and potential of our proposed approach in advancing the field of infrared object detection."

  1. The image is not clear when enlarged.

Response: Thank you for your suggestion. We have reuploaded clearer images.

Reviewer 3 Report

The paper is well written, but the following points need to be addressed.

1)      Rewrite the abstract in more informative manner. Specify what types of challenges faced by current vehicle detection techniques. Clear explain the motivation both in abstract and introduction section. The conclusion is also very short. Elaborate.

2)      The citations to the compared methods are missing. For example, YOLOv4-mobilenet, YOLOv4-tiny, YOLOv5s, and YOLOv7-tiny.

3)      Briefly explain the mechanisms of YOLOv4-mobilenet, YOLOv4-tiny, YOLOv5s, and YOLOv7-tiny methods. How these are different from the proposed method is unclear.

4)      You have mentioned that,The use of an Anchor Free detection head instead of an Anchor Based detection head is employed to enhance detection speed.”  But the difference between anchor free and anchor based is not  properly analysed. Provide some experimental results to validate this.

5)      Replace the name of Section 4. Here, “Experiment” looks incomplete. You can write “Experimental Results and discussion”.

6)      What is the source of the dataset used? You have created? If not then cite appropriately.

7)      Include the visual results of more images (Atleast 4/5), clearly indicating the source image.

8)      All the references are not uniform. Check the MDPI reference style and modify accordingly. Some have month, some do not have. Employ similar structure for all the journals ans same for all the conferences.

9)      If possible perform some ablation study to defend the component selection of your method.

10)   Include some more recent published articles to your manuscript.

11)   Check the whole manuscript for any grammatical, spelling, punctuation and format errors.

The English Language is fine. Minor editing is required.

Author Response

Dear Editor/Reviewer,

Thank you for reviewing our previous submission and providing valuable feedback and suggestions. We are hereby submitting the revised manuscript, along with a detailed response addressing each of the review comments and outlining the modifications we have made.

Firstly, we sincerely appreciate your recognition of our research topic and objectives, as well as your encouragement and support for our work. In the revised manuscript, we have carefully considered each of the comments you raised and made the necessary revisions and improvements accordingly. Please refer to the "Responses" section for the specific details of the modifications, and we have highlighted the revised portions in the revised manuscript.

Lastly, we would like to express our gratitude once again for your attention and review of our research. We deeply appreciate the valuable insights you have provided, as they have played a crucial role in enhancing the quality and academic rigor of our study. We hope that you will be satisfied with our revised manuscript and consider it for publication in your esteemed journal.

Best regards,

Zixuan Zhang

[email protected]

  • Rewrite the abstract in more informative manner. Specify what types of challenges faced by current vehicle detection techniques. Clear explain the motivation both in abstract and introduction section. The conclusion is also very short. Elaborate.

Response: We deeply appreciate the reviewer’s suggestion. We have added descriptions of challenges and motivation in both the abstract and introduction sections. Additionally, we have included necessary information in the conclusion.

“Abstract: In the field of object detection algorithms, the task of infrared vehicle detection holds significant importance. By utilizing infrared sensors, this approach detects the thermal radiation emitted by vehicles, enabling robust vehicle detection even during nighttime or adverse weather conditions, thus enhancing traffic safety and the efficiency of intelligent driving systems. Current techniques for infrared vehicle detection encounter difficulties in handling low contrast, detecting small objects, and ensuring real-time performance. In the domain of lightweight object detection algorithms, certain existing methodologies face challenges in effectively balancing detection speed and accuracy for this specific task. In order to address this quandary, ......”

“Introduction: ...... However, current infrared vehicle detection technology faces several challenges within the domain of object detection algorithms, including issues related to low contrast, small object detection, and real-time performance. Some existing lightweight target detection algorithms struggle to effectively balance detection speed and accuracy in this particular task. In response to this challenge, researchers are actively exploring novel techniques and methodologies. One potential solution involves the integration of deep learning with conventional computer vision technologies to overcome the challenges associated with low contrast and small object detection. Furthermore, enhancing real-time performance is of paramount importance, particularly in the context of autonomous driving systems, as it necessitates rapid and precise detection of surrounding vehicles and obstacles....”

“5. Conclusion

In conclusion, this paper has presented YOLO-IR-Free, an advanced algorithm tailored for the real-time detection of vehicles in infrared imagery. The motivation behind this work stemmed from the critical need to enhance traffic safety and the efficiency of intelligent driving systems by addressing the challenges associated with infrared vehicle detection. These challenges include low contrast, the detection of small objects, and the imperative requirement for real-time performance.

To overcome these challenges, we introduced a novel attention mechanism and network module into the YOLOv7 framework, effectively empowering our algorithm to capture subtle textures and low-contrast features inherent in infrared images. Notably, our decision to employ an Anchor-Free detection head, as opposed to the traditional Anchor-Based approach, significantly improved detection speed without compromising accuracy. Through extensive experiments, YOLO-IR-Free demonstrated superior performance metrics, including accuracy, recall rate, and F1 score, all while maintaining impeccable real-time capabilities.

Looking ahead, the future holds exciting prospects for YOLO-IR-Free. We envision further optimizations to enhance its overall performance and real-time capabilities. Additionally, we anticipate extending the application of our algorithm to tackle a broader spectrum of infrared object detection tasks. As intelligent driving systems continue to evolve, YOLO-IR-Free stands as a promising solution for safer and more efficient roadways, underpinning the potential for groundbreaking advancements in the field of infrared object detection.”

  • The citations to the compared methods are missing. For example, YOLOv4-mobilenet, YOLOv4-tiny, YOLOv5s, and YOLOv7-tiny.

Response: Thank you for your comments. In Table 1, we compared the performance of Faster-RCNN, YOLOv4-mobilenet, YOLOv4-Tiny, YOLOv5s, and YOLOv7-tiny. Additionally, we have included a comparison with YOLOv8s.

"To demonstrate the performance advantages of YOLO-IR-Free, we compare it with the original YOLOv7 and other improved algorithms. Table 1 presents the performance comparison between our YOLO-IR-Free algorithm and Faster-RCNN, YOLOv4-mobilenet, YOLOv4-tiny, YOLOv5s, and YOLOv7-tiny. In terms of precision, our YOLO-IR-Free achieves the highest value of 0.95 among all algorithms. The second and third-ranked algorithms are YOLOv5s with a precision of 0.925 and YOLOv7-tiny with a precision of 0.923. For recall, our YOLO-IR-Free achieves the highest value of 0.902, which is also the best among all compared algorithms. In terms of F1 score, YOLO-IR-Free achieves 0.92, while YOLOv7-tiny and YOLOv5s score 0.88 and 0.89, respectively. Furthermore, YOLO-IR-Free exhibits outstanding real-time capability, reaching a frame rate of 192 FPS, far surpassing Faster-RCNN, YOLOv4-mobilenet, and YOLOv5s. The YOLOv7-tiny algorithm achieves a higher Frames Per Second (FPS) compared to YOLOv8s, yet exhibits lower values across key performance metrics including Precision, Recall, F1, and mean Average Precision (mAP) in comparison to YOLOv8s. In contrast, our YOLO-IR-Free algorithm demonstrates notable improvements over YOLOv7-tiny, boasting a slight advantage over YOLOv8s in terms of Precision, Recall, and mAP. Moreover, it holds a significant edge in terms of FPS. Despite YOLOv4-tiny having the highest FPS, its detection accuracy is significantly lower."

  • Briefly explain the mechanisms of YOLOv4-mobilenet, YOLOv4-tiny, YOLOv5s, and YOLOv7-tiny methods. How these are different from the proposed method is unclear.

Response: Thank you for the suggestion. In Section 3.4, we have provided explanations for YOLOv4-mobilenet, YOLOv4-tiny, YOLOv5s, and YOLOv7-tiny, along with highlighting the distinctions of the proposed method.

“Before introducing our YOLO-IR-FREE algorithm, it is imperative to acquaint oneself with several other lightweight object detection algorithms. YOLOv4-tiny, a streamlined variant of YOLOv4, merits particular attention due to its parsimonious parameterization, merely encompassing 6 million parameters, equivalent to one-tenth of its precursor, which leads to a substantial acceleration in detection speed. Comprising a network architecture with a total depth of 38 layers, it incorporates three residual units, LeakyReLU activation functions, and a shift in target classification and regression to two distinct feature layers, synergistically leveraging a Feature Pyramid Network (FPN). YOLOv4-mobilenet, an offspring of YOLOv4, diverges by adopting the lighter Mobilenet backbone. YOLOv5S, as developed by Glenn Jocher, represents another noteworthy lightweight object detection algorithm. In contrast to YOLOv4, YOLOv5 introduces an array of enhancements, including an enriched data augmentation regimen and Weighted NMS, among others. YOLOv7-Tiny, designed expressly for resource-constrained devices, stands as an efficient lightweight object detection algorithm. Building upon the foundation of YOLOv7, it achieves real-time operation on embedded and mobile platforms through a more compact network architecture and optimized training strategies, thereby reducing model parameters and computational requirements.

Figure 4 presents our proposed YOLO-IR-Free algorithm, which consists of the Anchor Free Head, TSE feature fusion mechanism module, and Rep-LAN network structure. The CBS module is the Convolution layer, Batch Normalization layer and SILU activation layer. The MP module is the Maxpooling layer. The SILU activation function is utilized in the YOLO-IR-Free algorithm. Relative to several other lightweight object detection algorithms, our YOLO-IR-Free algorithm demonstrates a distinct advantage in the domain of infrared vehicle detection. Infrared vehicle images often present challenges such as occlusion and low contrast. Our TSE (Thermal Spatial Enhancement) attention mechanism enhances the focus on vehicle-related features, mitigating these challenges. Furthermore, the Rep-LAN (Receptive Field Pyramidal Long-range Attention Network) architecture amalgamates the computational efficiency of Rep-Conv (Receptive Field Pyramidal Convolution) with the benefits of short-path multi-block stacking, thereby augmenting the network's feature extraction capabilities.”

  • You have mentioned that “The use of an Anchor Free detection head instead of an Anchor Based detection head is employed to enhance detection speed.” But the difference between anchor free and anchor based is not properly analyzed. Provide some experimental results to validate this.

Response: Thank you for your careful review. We have added comparative experiments between anchor-free and anchor-based methods in the experimental section.

“In order to validate the impact of the Anchor-Free detection head on network performance, we initially replaced the detection head of YOLO-IR-Free with a conventional Anchor-based detection head for experimentation. Table 2 presents the performance comparison upon removal of the Anchor-Free detection head. It is observed that Precision, Recall, and mAP experience slight decreases, whereas FPS exhibits a significant decline. The experimental results underscore the enhancements brought about by the Anchor-Free detection head across various performance metrics, notably a substantial improvement in inference speed.”

  • Replace the name of Section 4. Here, “Experiment” looks incomplete. You can write “Experimental Results and discussion”.

Response: Thank you for your careful review. We have changed the title of Section 4 to "Experimental Results and Discussion."

  • What is the source of the dataset used? You have created? If not then cite appropriately.

Response: Thank you for your suggestion. The dataset we utilized was collected and curated by ourselves, sourced from the Internet or extracted and organized from other datasets. We can provide appropriate references to assist readers who are unable to verify or compare the results. “FREE Teledyne FLIR Thermal Dataset: https://www.flir.com/oem/adas/adas-dataset-form/”

  • Include the visual results of more images (at least 4/5), clearly indicating the source image.

Response: Thank you for the suggestion. We have uploaded higher resolution images.

  • All the references are not uniform. Check the MDPI reference style and modify it accordingly. Some have month, some do not have. Employ similar structure for all the journals and same for all the conferences.

Response: Thank you for your careful review. We have standardized the format of the references.

  • If possible perform some ablation study to defend the component selection of your method.

Response: Thank you for your suggestion. In Section 4.3 of the experiment, we have added an ablation study part, analyzing the performance improvement of TSE, Anchor-Free detection head, and Rep-LAN components.

“In order to validate the impact of the Anchor-Free detection head on network performance, we initially replaced the detection head of YOLO-IR-Free with a conventional Anchor-based detection head for experimentation. Table 2 presents the performance comparison upon removal of the Anchor-Free detection head. It is observed that Precision, Recall, and mAP experience slight decreases, whereas FPS exhibits a significant decline. The experimental results underscore the enhancements brought about by the Anchor-Free detection head across various performance metrics, notably a substantial improvement in inference speed.”

  • Include some more recent published articles to your manuscript.

Response: Thank you for your suggestion. We have added some references from the past two years.

  • Check the whole manuscript for any grammatical, spelling, punctuation and format errors.

Response: Thank you for your careful review. We have thoroughly reviewed the article for grammatical, spelling, punctuation, and formatting errors.

Reviewer 4 Report

In this paper, the authors put forward a new attention mechanism and network module to effectively capture subtle textures and low-contrast features in infrared images. The authors deserve credit for their novel approach to addressing challenges in infrared vehicle detection. However, there are certain aspects of the paper that need refinement to bolster its credibility and clarity.

1. Literature Review: It is crucial to provide a comprehensive and well-documented survey of the existing literature on infrared vehicle detection. The paper appears to rely heavily on citations from review papers, neglecting detailed introductions to relevant vehicle detection works. A more exhaustive literature review, with direct comparisons and distinctions from prior research, would significantly strengthen the paper's foundation.

2. Comparison with YOLOv8: The authors claim to have implemented an Anchor-Free approach through modifications to the YOLOv7 network module. Given that YOLOv8 is a recognized Anchor-Free algorithm, it is essential to compare the proposed Anchor-Free implementation in YOLO-IR-Free with the established YOLOv8. This comparison would elucidate the advancements made and help establish the uniqueness of the proposed algorithm.

3. Network Design: While the paper emphasizes the use of infrared images for vehicle detection, it lacks specificity in describing how the network architecture is tailored to infrared vehicle detection. Providing insights into design choices that account for infrared-specific challenges could enhance the paper's contribution and relevance.

4. Ablation Experiments: To validate the effectiveness of the proposed approach, it is imperative to include ablation experiments. These experiments would help dissect the contributions of individual components (such as the attention mechanism, network module, and Anchor-Free detection head) to the overall performance improvement. A detailed analysis of these experiments would add depth to the paper's findings.

In conclusion, "YOLO-IR-Free: An Improved Algorithm for Real-time Detection of Vehicles in Infrared Images" presents a promising direction for addressing challenges in infrared vehicle detection. However, to enhance the paper's impact and credibility, the authors should address the issues highlighted above. By refining the literature review, conducting a comparative analysis with YOLOv8, elaborating on infrared-specific network design, and including ablation experiments, the paper can make a more compelling case for the effectiveness of the proposed YOLO-IR-Free algorithm.

Overall, there's nothing fundamentally wrong with it; it just requires a bit more attention to detail.

Author Response

Dear Editor/Reviewer,

Thank you for reviewing our previous submission and providing valuable feedback and suggestions. We are hereby submitting the revised manuscript, along with a detailed response addressing each of the review comments and outlining the modifications we have made.

Firstly, we sincerely appreciate your recognition of our research topic and objectives, as well as your encouragement and support for our work. In the revised manuscript, we have carefully considered each of the comments you raised and made the necessary revisions and improvements accordingly. Please refer to the "Responses" section for the specific details of the modifications, and we have highlighted the revised portions in the revised manuscript.

Lastly, we would like to express our gratitude once again for your attention and review of our research. We deeply appreciate the valuable insights you have provided, as they have played a crucial role in enhancing the quality and academic rigor of our study. We hope that you will be satisfied with our revised manuscript and consider it for publication in your esteemed journal.

Best regards,

Zixuan Zhang

[email protected]

  1. Literature Review: It is crucial to provide a comprehensive and well-documented survey of the existing literature on infrared vehicle detection. The paper appears to rely heavily on citations from review papers, neglecting detailed introductions to relevant vehicle detection works. A more exhaustive literature review, with direct comparisons and distinctions from prior research, would significantly strengthen the paper's foundation.

Response: Thank you for your careful review. We have added introductions and comparisons of some works related to infrared vehicle detection in the "Related Work" section.

“In 2022, Jiang et al. proposes a framework for object detection in UAV thermal infrared (TIR) images and videos using YOLO models. The study utilizes convolutional neural network (CNN) architecture to extract features from TIR data captured by FLIR cameras. The results show high mean average precision (mAP) and fast detection speed with the YOLOv5-s model. It provides valuable insights into the qualitative and quantitative evaluation of object detection from TIR images and videos using deep learning models. In 2021, Du et al. presents a new approach to detect weak and occluded vehicles in complex infrared environments using an improved YOLOv4 model. The proposed method involvInies secondary transfer learning from visible dataset to infrared dataset, and the addition of a hard negative example mining block to the YOLOv4 model. The results show that the improved model achieved a detection accuracy of 91.59%, indicating its potential for real-world applications in surveillance and security. In 2021, Liu presents a robust thermal infrared vehicle and pedestrian detection method for accurately detecting motion-blurred, tiny, and dense objects in complex scenes. The method proposes an optimized feature selective anchor-free (FSAF) module with a weight parameter β, enhancing the detection performance of motion-blurred objects and improving detection precision for tiny and dense objects when combined with the YOLOv3 single-shot detector. Experimental results show that the proposed method outperforms other thermal infrared detection algorithms, achieving a mean average precision (mAP) of 72.2%. Existing methods for infrared image-based vehicle detection have made certain achievements in accuracy, but still have limitations. For example, some methods are limited in handling complex scenes, small objects, and low contrast, making them unable to meet the real-time detection requirements of edge computing platforms. On the other hand, while the YOLO series methods perform well in real-time applications, their accuracy and recall rates may be slightly inferior to other methods in certain cases. To address these issues, this paper proposes an improved method for infrared image-based vehicle detection to enhance detection performance.”

  1. Comparison with YOLOv8: The authors claim to have implemented an Anchor-Free approach through modifications to the YOLOv7 network module. Given that YOLOv8 is a recognized Anchor-Free algorithm, it is essential to compare the proposed Anchor-Free implementation in YOLO-IR-Free with the established YOLOv8. This comparison would elucidate the advancements made and help establish the uniqueness of the proposed algorithm.

Response: We deeply appreciate the reviewer’s suggestion. In Section 4.2, we have added comparative experimental analysis with YOLOv8s.

"The YOLOv7-tiny algorithm achieves a higher Frames Per Second (FPS) compared to YOLOv8s, yet exhibits lower values across key performance metrics including Precision, Recall, F1, and mean Average Precision (mAP) in comparison to YOLOv8s. In contrast, our YOLO-IR-Free algorithm demonstrates notable improvements over YOLOv7-tiny, boasting a slight advantage over YOLOv8s in terms of Precision, Recall, and mAP. Moreover, it holds a significant edge in terms of FPS."

  1. Network Design: While the paper emphasizes the use of infrared images for vehicle detection, it lacks specificity in describing how the network architecture is tailored to infrared vehicle detection. Providing insights into design choices that account for infrared-specific challenges could enhance the paper's contribution and relevance.

Pesponse: Thank you for your suggestion. In Section 3.4, we have added an analysis of how the proposed network module design helps address the challenges in infrared vehicle detection.

"Relative to several other lightweight object detection algorithms, our YOLO-IR-Free algorithm demonstrates a distinct advantage in the domain of infrared vehicle detection. Infrared vehicle images often present challenges such as occlusion and low contrast. Our TSE (Thermal Spatial Enhancement) attention mechanism enhances the focus on vehicle-related features, mitigating these challenges. Furthermore, the Rep-LAN (Receptive Field Pyramidal Long-range Attention Network) architecture amalgamates the computational efficiency of Rep-Conv (Receptive Field Pyramidal Convolution) with the benefits of short-path multi-block stacking, thereby augmenting the network's feature extraction capabilities."

  1. Ablation Experiments: To validate the effectiveness of the proposed approach, it is imperative to include ablation experiments. These experiments would help dissect the contributions of individual components (such as the attention mechanism, network module, and Anchor-Free detection head) to the overall performance improvement. A detailed analysis of these experiments would add depth to the paper's findings.

Response: Thank you for your suggestion. In Section 4.3, we have added an analysis of the ablation experiments on the TSE attention mechanism, Rep-LAN, and anchor-free methods.

"4.3 Ablation Experiment

In order to validate the impact of the Anchor-Free detection head on network performance, we initially replaced the detection head of YOLO-IR-Free with a conventional Anchor-based detection head for experimentation. Table 2 presents the performance comparison upon removal of the Anchor-Free detection head. It is observed that Precision, Recall, and mAP experience slight decreases, whereas FPS exhibits a significant decline. The experimental results underscore the enhancements brought about by the Anchor-Free detection head across various performance metrics, notably a substantial improvement in inference speed."

Round 2

Reviewer 1 Report

The author failed to justify their approach. The existing frameworks are well-developed, have proven accuracy, and tested in existing benchmarks.

Author Response

It is recommended that reviewers read the paper carefully when reviewing the manuscript and give practical and feasible suggestions. We only hope to get fair, reliable and serious review results.

Reviewer 3 Report

Authors have addressed all but one of my comments. They missed only but important comment which need to be resolved before acceptance. Please add visual results of  3/4 more images. For example, the recent updated manuscript shows the visual result for only a single image which is presented in Fig 11. Like this show results for 3/4 more images. You may select images that are shown in Fig. 7 (You have shown 9 images here but shown the results for only a single image. This fails to convince the workability of the proposed method. include results for 3/4 more images from Fig. 7) 

NA. 

Author Response

Dear Editor/Reviewer,

Thank you for reviewing our previous submission and providing valuable feedback and suggestions. We are hereby submitting the revised manuscript, along with a detailed response addressing each of the review comments and outlining the modifications we have made.

With kind regards,

Zixuan Zhang

[email protected]

Respones

Reviewer 3:

Authors have addressed all but one of my comments. They missed only but important comment which need to be resolved before acceptance. Please add visual results of  3/4 more images. For example, the recent updated manuscript shows the visual result for only a single image which is presented in Fig 11. Like this show results for 3/4 more images. You may select images that are shown in Fig. 7 (You have shown 9 images here but shown the results for only a single image. This fails to convince the workability of the proposed method. include results for 3/4 more images from Fig. 7)

Respones: Thank you for your suggestion. In Fig. 11, we added a comparison of the 9 images displayed in Fig. 7. Our algorithm has fewer missed and false detections than other algorithms.

Reviewer 4 Report

The result obtained by the proposed network is only marginally better than yolov8, which raises doubts about its superiority over the existing method. It would be helpful if the authors could share their code on a public platform.

Author Response

Dear Editor/Reviewer,

Thank you for reviewing our previous submission and providing valuable feedback and suggestions. We are hereby submitting the revised manuscript, along with a detailed response addressing each of the review comments and outlining the modifications we have made.

With kind regards,

Zixuan Zhang

[email protected]

Respones

Reviewer 4:

The result obtained by the proposed network is only marginally better than yolov8, which raises doubts about its superiority over the existing method. It would be helpful if the authors could share their code on a public platform.

Respones: Thank you for your suggestion. We concur with your point, and we are currently considering sharing our code and weight files after the publication.  
